# Learning From Noisy Singly-labeled Data

**Ashish Khetan**
University of Illinois at Urbana-Champaign
Urbana, IL 61801
khetan2@illinois.edu

**Zachary C. Lipton**
Amazon Web Services
Seattle, WA 98101
liptoz@amazon.com

**Animashree Anandkumar**
Amazon Web Services
Seattle, WA 98101
anima@amazon.com

## Abstract

Supervised learning depends on annotated examples, which are taken to be the *ground truth*. But these labels often come from noisy crowdsourcing platforms, like Amazon Mechanical Turk. Practitioners typically collect multiple labels per example and aggregate the results to mitigate noise (the classic crowdsourcing problem). Given a fixed annotation budget and unlimited unlabeled data, redundant annotation comes at the expense of fewer labeled examples. This raises two fundamental questions: (1) How can we best learn from noisy workers? (2) How should we allocate our labeling budget to maximize the performance of a classifier? We propose a new algorithm for jointly modeling labels and worker quality from noisy crowd-sourced data. The alternating minimization proceeds in rounds, estimating worker quality from disagreement with the current model and then updating the model by optimizing a loss function that accounts for the current estimate of worker quality. Unlike previous approaches, even with only one annotation per example, our algorithm can estimate worker quality. We establish a generalization error bound for models learned with our algorithm and establish theoretically that it's better to label many examples once (vs less multiply) when worker quality exceeds a threshold. Experiments conducted on both ImageNet (with simulated noisy workers) and MS-COCO (using the real crowdsourced labels) confirm our algorithm's benefits.

## 1 Introduction

Recent advances in supervised learning owe, in part, to the availability of large annotated datasets. For instance, the performance of modern image classifiers saturates only with millions of labeled examples. This poses an economic problem: Assembling such datasets typically requires the labor of human annotators. If we confined the labor pool to experts, this work might be prohibitively expensive. Therefore, most practitioners turn to crowdsourcing platforms such as Amazon Mechanical Turk (AMT), which connect employers with low-skilled workers who perform simple tasks, such as classifying images, at low cost.

Compared to experts, crowd-workers provide noisier annotations, possibly owing to high variation in worker skill; and a per-answer compensation structure that encourages rapid answers, even at the expense of accuracy. To address variation in worker skill, practitioners typically collect multiple independent labels for each training example from different workers. In practice, these labels are often aggregated by applying a simple majority vote. Academics have proposed many efficient algorithms for estimating the ground truth from noisy annotations. Research addressing the crowd-sourcing problem goes back to the early 1970s. Dawid & Skene (1979) proposed a probabilistic model to jointly estimate worker skills and ground truth labels and used expectation maximization (EM) to estimate the parameters. Whitehill et al. (2009); Welinder et al. (2010); Zhou et al. (2015) proposed generalizations of the Dawid-Skene model, e.g. by estimating the difficulty of each example.

Although the downstream goal of many crowdsourcing projects is to train supervised learning models, research in the two disciplines tends to proceed in isolation. Crowdsourcing research seldom accounts for the downstream utility of the produced annotations as training data in machine learning (ML) algorithms. And ML research seldom exploits the noisy labels collected from multiple human workers. A few recent papers use the original noisy labels and the corresponding worker identities together with the predictions of a supervised learning model trained on those same labels, to estimate the ground truth (Branson et al., 2017; Guan et al., 2017; Welinder et al., 2010). However, these papers do not realize the full potential of combining modeling and crowd-sourcing. In particular, they are unable to estimate worker qualities when there is only one label per training example.

**This paper presents** a new supervised learning algorithm that alternately models the labels and worker quality. The EM algorithm bootstraps itself in the following way: Given a trained model, the algorithm estimates worker qualities using the disagreement between workers and the current predictions of the learning algorithm. Given estimated worker qualities, our algorithm optimizes a suitably modified loss function. We show that accurate estimates of worker quality can be obtained even when only collecting one label per example provided that each worker labels sufficiently many examples. An accurate estimate of the worker qualities leads to learning a better model. This addresses a shortcoming of the prior work and overcomes a significant hurdle to achieving practical crowdsourcing without redundancy.

We give theoretical guarantees on the performance of our algorithm. We analyze the two alternating steps: (a) estimating worker qualities from disagreement with the model, (b) learning a model by optimizing the modified loss function. We obtain a bound on the accuracy of the estimated worker qualities and the generalization error of the model. Through the generalization error bound, we establish that it is better to label many examples once than to label less examples multiply when worker quality is above a threshold. Empirically, we verify our approach on several multi-class classification datasets: ImageNet and CIFAR10 (with simulated noisy workers), and MS-COCO (using the real noisy annotator labels). Our experiments validate that when the cost of obtaining unlabeled examples is negligible and the total annotation budget is fixed, it is best to collect a single label per training example for as many examples as possible. We emphasize that although this paper applies our approach to classification problems, the main ideas of the algorithm can be extended to other tasks in supervised learning.

## 2 RELATED WORK

The traditional crowdsourcing problem addresses the challenge of aggregating multiple noisy labels. A naive approach is to aggregate the labels based on majority voting. More sophisticated agreement-based algorithms jointly model worker skills and ground truth labels, estimating both using EM or similar techniques (Dawid & Skene, 1979; Jin & Ghahramani, 2003; Whitehill et al., 2009; Welinder et al., 2010; Zhou et al., 2012; Liu et al., 2012; Dalvi et al., 2013; Liu et al., 2012). Zhang et al. (2014) shows that the EM algorithm with spectral initialization achieves minimax optimal performance under the Dawid-Skene model. Karger et al. (2014) introduces a message-passing algorithm for estimating binary labels under the Dawid-Skene model, showing that it performs strictly better than majority voting when the number of labels per example exceeds some threshold. Similar observations are made by (Bragg et al., 2016). A primary criticism of EM-based approaches is that in practice, it's rare to collect more than $3$ to $5$ labels per example; and with so little redundancy, the small gains achieved by EM over majority voting are not compelling to practitioners. In contrast, our algorithm performs well in the low-redundancy setting. Even with just one label per example, we can accurately estimate worker quality.

Several prior crowdsourcing papers incorporate the predictions of a supervised learning model, together with the noisy labels, to estimate the ground truth labels. Welinder et al. (2010) consider binary classification and frames the problem as a generative Bayesian model on the features of the examples and the labels. Branson et al. (2017) consider a generalization of the Dawid-Skene model and estimate its parameters using supervised learning in the loop. In particular, they consider a joint probability over observed image features, ground truth labels, and the worker labels and compute the maximum likelihood estimate of the true labels using alternating minimization. We also consider a joint probability model but it is significantly different from theirs as we assume that the optimal labeling function gives the ground truth labels. We maximize the joint likelihood using a variation

of expectation maximization to learn the optimal labeling function and the true labels. Further, they train the supervised learning model using the intermediate predictions of the labels whereas we train the model by minimizing a weighted loss function where the weights are the intermediate posterior probability distribution of the labels. Moreover, with only one label per example, their algorithm fails and estimates all the workers to be equally good. They only consider binary classification, whereas we verify our algorithm on multi-class (ten classes) classification problem.

A rich body of work addresses human-in-loop annotation for computer vision tasks. However, these works assume that humans are experts, i.e., that they give noiseless annotations (Branson et al., 2010; Deng et al., 2013; Wah et al., 2011). We assume workers are unreliable and have varying skills. A recent work by Ratner et al. (2016) also proposes to use predictions of a supervised learning model to estimate the ground truth. However, their algorithm is significantly different than ours as it does not use iterative estimation technique, and their approach of incorporating worker quality parameters in the supervised learning model is different. Their theoretical results are limited to the linear classifiers.

Another line of work employs active learning, iteratively filtering out examples for which aggregated labels have high confidence and collect additional labels for the remaining examples (Whitehill et al., 2009; Welinder & Perona, 2010; Khetan & Oh, 2016). The underlying modeling assumption in these papers is that the questions have varying levels of difficulty. At each iteration, these approaches employ an EM-based algorithm to estimate the ground truth label of the remaining unclassified examples. For simplicity, our paper does not address example difficulties, but we could easily extend our model and algorithm to accommodate this complexity.

Several papers analyze whether repeated labeling is useful. Sheng et al. (2008) analyzed the effect of repeated labeling and showed that it depends upon the relative cost of getting an unlabeled example and the cost of labeling. Ipeirotis et al. (2014) shows that if worker quality is below a threshold then repeated labeling is useful, otherwise not. Lin et al. (2014a; 2016) argues that it also depends upon expressiveness of the classifier in addition to the factors considered by others. However, these works do not exploit predictions of the supervised learning algorithm to estimate the ground truth labels, and hence their findings do not extend to our methodology.

Another body of work that is relevant to our problem is learning with noisy labels where usual assumption is that all the labels are generated through the same noisy rate given their ground truth label. Recently Natarajan et al. (2013) proposed a generic unbiased loss function for binary classification with noisy labels. They employed a modified loss function that can be expressed as a weighted sum of the original loss function, and gave theoretical bounds on the performance. However, their weights become unstably large when the noise rate is large, and hence the weights need to be tuned. Sukhbaatar et al. (2014); Jindal et al. (2016) learns noise rate as parameters of the model. A recent work by Guan et al. (2017) trains an individual softmax layer for each expert and then predicts their weighted sum where weights are also learned by the model. It is not scalable to crowdsourcing scenario where there are thousands of workers. There are works that aim to create noise-robust models (Joulin et al., 2016; Krause et al., 2016), but they are not relevant to our work.

## 3 PROBLEM FORMULATION

Let $\mathcal{D}$ be the underlying true distribution generating pairs $(X, Y) \in \mathcal{X} \times \mathcal{K}$ from which $n$ i.i.d. samples $(X_1, Y_1), (X_2, Y_2), \cdots, (X_n, Y_n)$ are drawn, where $\mathcal{K}$ denotes the set of possible labels $\mathcal{K} := \{1, 2, \cdots, K\}$, and $\mathcal{X} \subseteq \mathbb{R}^d$ denotes the set of euclidean features. We denote the marginal distribution of $Y$ by $\{q_1, q_2, \cdots, q_K\}$, which is unknown to us. Consider a pool of $m$ workers indexed by $1, 2, \cdots, m$. We use $[m]$ to denote the set $\{1, 2, \cdots, m\}$. For each $i$-th sample $X_i$, $r$ workers $\{w_{ij}\}_{j \in [r]} \in [m]^r$ are selected randomly, independent of the sample $X_i$. Each selected worker provides a noisy label $Z_{ij}$ for the sample $X_i$, where the distribution of $Z_{ij}$ depends on the selected worker and the true label $Y_i$. We call $r$ the *redundancy* and, for simplicity, assume it to be the same for each sample. However, our algorithm can also be applied when redundancy varies across the samples. We use $Z_i^{(r)}$ to denote $\{Z_{ij}\}_{j \in [r]}$, the set of $r$ labels collected on the $i$-th example, and $w_i^{(r)}$ to denote $\{w_{ij}\}_{j \in [r]}$.

Following Dawid & Skene (1979), we assume the probability that the $a$-th worker labels an item in class $k \in \mathcal{K}$ as class $s \in \mathcal{K}$ is independent of any particular chosen item, that is, it is a constant over

$i \in [n]$. Let us denote this constant by $\pi_{ks}$; by definition, $\sum_{s \in \mathcal{K}} \pi_{ks} = 1$ for all $k \in \mathcal{K}$, and we call $\pi^{(a)} \in [0,1]^{K \times K}$ the confusion matrix of the $a$-th worker. In particular, the distribution of $Z$ is:

$$\mathbb{P}\left[Z_{ij} = s \mid Y_i = k, w_{ij} = a\right] = \pi_{ks}^{(a)} . \tag{1}$$

The diagonal entries of the confusion matrix correspond to the probabilities of correctly labeling an example. The off-diagonal entries represent the probability of mislabeling. We use $\pi$ to denote the collection of confusion matrices $\{\pi^{(a)}\}_{a \in [m]}$.

We assume $nr$ workers $w_{1,1}, w_{1,2}, \cdots, w_{n,r}$ are selected uniformly at random from a pool of $m$ workers with replacement and a batch of $r$ workers are assigned to each of the examples $X_1, X_2, \cdots, X_n$. The corrupted labels along with the worker information $(X_1, Z_1^{(r)}, w_1^{(r)}), \cdots, (X_n, Z_n^{(r)}, w_n^{(r)})$ are what the learning algorithm sees.

Let $\mathcal{F}$ be the hypothesis class, and $f \in \mathcal{F}$, $f : \mathcal{X} \to \mathbb{R}^K$, denote a vector valued predictor function. Let $\ell(f(X), Y)$ denote a loss function. For a predictor $f$, its $\ell$-risk under $\mathcal{D}$ is defined as

$$R_{\ell, \mathcal{D}}(f) \ := \ \mathbb{E}_{(X,Y) \sim \mathcal{D}}\left[\ell(f(X), Y)\right] . \tag{2}$$

Given the observed samples $(X_1, Z_1^{(r)}, w_1^{(r)}), \cdots, (X_n, Z_n^{(r)}, w_n^{(r)})$, we want to learn a good predictor function $\widehat{f} \in \mathcal{F}$ such that its risk under the true distribution $\mathcal{D}$, $R_{\ell, \mathcal{D}}(\widehat{f})$ is minimal. Having access to only noisy labels $Z^{(r)}$ by workers $w^{(r)}$, we compute $\widehat{f}$ as the one which minimizes a suitably modified loss function $\ell_{\widehat{\pi}, \widehat{q}}(f(X), Z^{(r)}, w^{(r)})$. Where $\widehat{\pi}$ denote an estimate of confusion matrix $\pi$, and $\widehat{q}$ an estimate of $q$, the prior distribution on $Y$. We define $\ell_{\widehat{\pi}, \widehat{q}}$ in the following section.

## 4 ALGORITHM

Assume that there exists a function $f^* \in \mathcal{F}$ such that $f^*(X_i) = Y_i$ for all $i \in [n]$. Under the Dawid-Skene model (described in previous section), the joint likelihood of true labeling function $f^*(X_i)$ and observed labels $\{Z_{ij}\}_{i \in [n], j \in [r]}$ as a function of confusion matrices of workers $\pi$ can be written as

$$L\left(\pi; f^*, \{X_i\}_{i \in [n]}, \{Z_{ij}\}_{i \in [n], j \in [r]}\right) :=$$
$$\prod_{i=1}^{n} \left(\sum_{k \in \mathcal{K}} q_k \mathbb{I}[f^*(X_i) = k] \left(\prod_{j=1}^{r} \left(\sum_{s \in \mathcal{K}} \mathbb{I}[Z_{ij} = s] \pi_{ks}^{(w_{ij})}\right)\right)\right) . \tag{3}$$

$q_k$'s are the marginal distribution of the true labels $Y_i$'s. We estimate the worker confusion matrices $\pi$ and the true labeling function $f^*$ by maximizing the likelihood function $L(\pi; f^*(X), Z)$. Observe that the likelihood function $L(\pi; f^*(X), Z)$ is different than the standard likelihood function of Dawid-Skene model in that we replace each true hidden labels $Y_i$ by $f^*(X_i)$. Like the EM algorithm introduced in (Dawid & Skene, 1979), we propose 'Model Bootstrapped EM' (MBEM) to estimate confusion matrices $\pi$ and the true labeling function $f^*$. EM converges to the true confusion matrices and the true labels given an appropriate spectral initialization of worker confusion matrices (Zhang et al., 2014). We show in Section 4.4 that MBEM converges under mild conditions when the worker quality is above a threshold and the number of training examples is sufficiently large. In the following two subsections, we motivate and explain our iterative algorithm to estimate the true labeling function $f^*$ given a good estimate of worker confusion matrices $\pi$ and vice-versa.

### 4.1 LEARNING WITH NOISY LABELS

To begin, we ask, *what is the optimal approach to learn the predictor function $\widehat{f}$ when for each worker we have $\widehat{\pi}$, a good estimation of the true confusion matrix $\pi$, and $\widehat{q}$, an estimate of the prior?* A recent paper, Natarajan et al. (2013) proposes minimizing an unbiased loss function specifically, a weighted sum of the original loss over each possible ground truth label. They provide weights for binary classification where each example is labeled by only one worker. Consider a worker with confusion matrix $\pi$, where $\pi_y > 1/2$ and $\pi_{-y} > 1/2$ represent her probability of correctly labeling the examples belonging to class $y$ and $-y$ respectively. Then their weights are $\pi_{-y}/(\pi_y + \pi_{-y} - 1)$ for class $y$ and $-(1 - \pi_y)/(\pi_y + \pi_{-y} - 1)$ for class $-y$. It is evident that their weights become

unstably large when the probabilities of correct classification $\pi_y$ and $\pi_{-y}$ are close to $1/2$, limiting the method's usefulness in practice. As explained below, for the same scenario, our weights would be $\pi_y/(1+\pi_y-\pi_{-y})$ for class $y$ and $(1-\pi_{-y})/(1+\pi_y-\pi_{-y})$ for class $-y$. Inspired by their idea, we propose weighing the loss function according to the posterior distribution of the true label given the $Z^{(r)}$ observed labels and an estimate of the confusion matrices of the worker who provided those labels. In particular, we define $\ell_{\widehat{\pi},\widehat{q}}$ to be

$$\ell_{\widehat{\pi},\widehat{q}}(f(X), Z^{(r)}, w^{(r)}) \;\; := \;\; \sum_{k \in \mathcal{K}} \mathbb{P}_{\widehat{\pi},\widehat{q}}[Y = k \mid Z^{(r)}; w^{(r)}] \, \ell(f(X), Y = k) \, . \tag{4}$$

If the observed label is uniformly random, then all weights are equal and the loss is identical for all predictor functions $f$. Absent noise, we recover the original loss function. Under the Dawid-Skene model, given the observed noisy labels $Z^{(r)}$, an estimate of confusion matrices $\widehat{\pi}$, and an estimate of prior $\widehat{q}$, the posterior distribution of the true labels can be computed as follows:

$$\mathbb{P}_{\widehat{\pi},\widehat{q}}[Y_i = k \mid Z_i^{(r)}; w_i^{(r)}] \;\; = \;\; \frac{\widehat{q}_k \prod_{j=1}^r \left( \sum_{s \in \mathcal{K}} \mathbb{I}[Z_{ij} = s]\widehat{\pi}_{ks}^{(w_{ij})} \right)}{\sum_{k' \in \mathcal{K}} \left( \widehat{q}_{k'} \prod_{j=1}^r \left( \sum_{s \in \mathcal{K}} \mathbb{I}[Z_{ij} = s]\widehat{\pi}_{k's}^{(w_{ij})} \right) \right)} \, , \tag{5}$$

where $\mathbb{I}[.]$ is the indicator function which takes value one if the identity inside it is true, otherwise zero. We give guarantees on the performance of the proposed loss function in Theorem 4.1. In practice, it is robust to noise level and significantly outperforms the unbiased loss function. Given $\ell_{\widehat{\pi},\widehat{q}}$, we learn the predictor function $\widehat{f}$ by minimizing the empirical risk

$$\widehat{f} \;\; \leftarrow \;\; \arg\min_{f \in \mathcal{F}} \frac{1}{n} \sum_{i=1}^n \ell_{\widehat{\pi},\widehat{q}}(f(X_i), Z_i^{(r)}, w_i^{(r)}) \, . \tag{6}$$

## 4.2 Estimating annotator noise

The next question is: how do we get a good estimate $\widehat{\pi}$ of the true confusion matrix $\pi$ for each worker. If redundancy $r$ is sufficiently large, we can employ the EM algorithm. However, in practical applications, redundancy is typically three or five. With so little redundancy, the standard applications of EM are of limited use. In this paper we look to transcend this problem, posing the question: Can we estimate confusion matrices of workers even when there is only one label per example? While this isn't possible in the standard approach, we can overcome this obstacle by incorporating a supervised learning model into the process of assessing worker quality.

Under the Dawid-Skene model, the EM algorithm estimates the ground truth labels and the confusion matrices in the following way: It alternately fixes the ground truth labels and the confusion matrices by their estimates and and updates its estimate of the other by maximizing the likelihood of the observed labels. The alternating maximization begins by initializing the ground truth labels with a majority vote. With only 1 label per example, EM estimates that all the workers are perfect.

We propose using model predictions as estimates of the ground truth labels. Our model is initially trained on the majority vote of the labels. In particular, if the model prediction is $\{t_i\}_{i \in [n]}$, where $t_i \in \mathcal{K}$, then the maximum likelihood estimate of confusion matrices and the prior distribution is given below. For the $a$-th worker, $\widehat{\pi}_{ks}^{(a)}$ for $k, s \in \mathcal{K}$, and $\widehat{q}_k$ for $k \in \mathcal{K}$, we have,

$$\widehat{\pi}_{ks}^{(a)} \;\; = \;\; \frac{\sum_{i=1}^n \sum_{j=1}^r \mathbb{I}[w_{ij} = a]\mathbb{I}[t_i = k]\mathbb{I}[Z_{ij} = s]}{\sum_{i=1}^n \sum_{j=1}^r \mathbb{I}[w_{ij} = a]\mathbb{I}[t_i = k]} \, , \qquad \widehat{q}_k = (1/n) \sum_{i=1}^n \mathbb{I}[t_i = k] \tag{7}$$

The estimate is effective when the hypothesis class $\mathcal{F}$ is expressive enough and the learner is robust to noise. Thus the model should, in general, have small training error on correctly labeled examples and large training error on wrongly labeled examples. Consider the case when there is only one label per example. The model will be trained on the raw noisy labels given by the workers. For simplicity, assume that each worker is either a *hammer* (always correct) or a *spammer* (chooses labels uniformly random). By comparing model predictions with the training labels, we can identify which workers are hammers and which are spammers, as long as each worker labels sufficiently many examples. We expect a hammer to agree with the model more often than a spammer.

### 4.3 ITERATIVE ALGORITHM

Building upon the previous two ideas, we present 'Model Bootstrapped EM', an iterative algorithm for efficient learning from noisy labels with small redundancy. MBEM takes data, noisy labels, and the corresponding worker IDs, and returns the best predictor function $\widehat{f}$ in the hypothesis class $\mathcal{F}$. In the first round, we compute the weights of the modified loss function $\ell_{\widehat{\pi},\widehat{q}}$ by using the weighted majority vote. Then we obtain an estimate of the worker confusion matrices $\widehat{\pi}$ using the maximum likelihood estimator by taking the model predictions as the ground truth labels. In the second round, weights of the loss function are computed as the posterior probability distribution of the ground truth labels conditioned on the noisy labels and the estimate of the confusion matrices obtained in the previous round. In our experiments, only two rounds are required to achieve substantial improvements over baselines.

---

**Algorithm 1** Model Bootstrapped EM (MBEM)

---

**Input:** $\{(X_i, Z_i^{(r)}, w_i^{(r)})\}_{i \in [n]}$, $T$ : number of iterations
**Output:** $\widehat{f}$ : predictor function
**Initialize posterior distribution using weighted majority vote**
 $\quad \mathbb{P}_{\widehat{\pi},\widehat{q}}[Y_i = k \mid Z_i^{(r)}; w_i^{(r)}] \leftarrow (1/r) \sum_{j=1}^{r} \mathbb{I}[Z_{ij} = k]$ , for $k \in \mathcal{K}, i \in [n]$
**Repeat $T$ times:**
 $\quad$**learn predictor function $\widehat{f}$**
 $\quad \widehat{f} \leftarrow \arg\min_{f \in \mathcal{F}} \frac{1}{n} \sum_{i=1}^{n} \sum_{k \in \mathcal{K}} \mathbb{P}_{\widehat{\pi},\widehat{q}}[Y_i = k \mid Z_i^{(r)}; w_i^{(r)}] \, \ell(f(X_i), Y_i = k)$
 $\quad$**predict on training examples**
 $\quad t_i \leftarrow \arg\max_{k \in \mathcal{K}} \widehat{f}(X_i)_k$, for $i \in [n]$
 $\quad$**estimate confusion matrices $\widehat{\pi}$ and prior class distribution $\widehat{q}$ given $\{t_i\}_{i \in [n]}$**
 $\quad \widehat{\pi}^{(a)} \leftarrow$ Equation (7), for $a \in [m]$; $\widehat{q} \leftarrow$ Equation (7)
 $\quad$**estimate label posterior distribution given $\widehat{\pi}, \widehat{q}$**
 $\quad \mathbb{P}_{\widehat{\pi},\widehat{q}}[Y_i = k \mid Z_i^{(r)}; w_i^{(r)}], \leftarrow$ Equation (5), for $k \in \mathcal{K}, i \in [n]$
**Return $\widehat{f}$**

---

### 4.4 PERFORMANCE GUARANTEES

The following result gives guarantee on the excess risk for the learned predictor function $\widehat{f}$ in terms of the VC dimension of the hypothesis class $\mathcal{F}$. Recall that risk of a function $f$ w.r.t. loss function $\ell$ is defined to be $R_{\ell,\mathcal{D}}(f) := \mathbb{E}_{(X,Y) \sim \mathcal{D}}[\ell(f(X), Y)]$, Equation (2). We assume that the classification problem is binary, and the distribution $q$, prior on ground truth labels $Y$, is uniform and is known to us. We give guarantees on the excess risk of the predictor function $\widehat{f}$, and accuracy of $\widehat{\pi}$ estimated in the second round. For the purpose of analysis, we assume that fresh samples are used in each round for computing function $\widehat{f}$ and estimating $\widehat{\pi}$. In other words, we assume that $\widehat{f}$ and $\widehat{\pi}$ are each computed using $n/4$ fresh samples in the first two rounds. We define $\alpha$ and $\beta_\epsilon$ to capture the average worker quality. Here, we give their concise bound for a special case when all the workers are identical, and their confusion matrix is represented by a single parameter, $0 \le \rho < 1/2$. Where $\pi_{kk} = 1 - \rho$, and $\pi_{ks} = \rho$ for $k \ne s$. Each worker makes a mistake with probability $\rho$. $\beta_\epsilon \le (\rho + \epsilon)^r \sum_{u=0}^{r} \binom{r}{u}(\tau^u + \tau^{r-u})^{-1}$, where $\tau := (\rho + \epsilon)/(1 - \rho - \epsilon)$. $\alpha$ for this special case is $\rho$. A general definition of $\alpha$ and $\beta_\epsilon$ for any confusion matrices $\pi$ is provided in the Appendix.

**Theorem 4.1.** *Define $N := nr$ to be the number of total annotations collected on $n$ training examples with redundancy $r$. Suppose $\min_{f \in \mathcal{F}} R_{\ell,\mathcal{D}}(f) \le 1/4$. For any hypothesis class $\mathcal{F}$ with a finite VC dimension $V$, and any $\delta < 1$, there exists a universal constant $C$ such that if $N$ is large enough and satisfies*

$$N \ge \max\left\{ Cr\left((\sqrt{V} + \sqrt{\log(1/\delta)})/(1 - 2\alpha)\right)^2, 2^{12} m \log(2^6 m/\delta) \right\}, \tag{8}$$

*then for binary classification with 0-1 loss function $\ell$, $\widehat{f}$ and $\widehat{\pi}$ returned by Algorithm 1 after $T = 2$ iterations satisfies*

$$R_{\ell,\mathcal{D}}(\widehat{f}) - \min_{f \in \mathcal{F}} R_{\ell,\mathcal{D}}(f) \le \frac{C\sqrt{r}}{1 - 2\beta_\epsilon}\left(\sqrt{\frac{V}{N}} + \sqrt{\frac{\log(1/\delta)}{N}}\right), \tag{9}$$

*and $\|\widehat{\pi}^{(a)} - \pi^{(a)}\|_\infty \le \epsilon_1$ for all $a \in [m]$, with probability at least $1 - \delta$. Where $\epsilon := 2^4\gamma + 2^8\sqrt{m\log(2^6 m\delta)/N}$, and $\gamma := \min_{f \in \mathcal{F}} R_{\ell,\mathcal{D}}(f) + C(\sqrt{V} + \sqrt{\log(1/\delta)})/((1-2\alpha)\sqrt{N/r})$. $\epsilon_1$ is defined to be $\epsilon$ with $\alpha$ in it replaced by $\beta_\epsilon$.*

The price we pay in generalization error bound on $\widehat{f}$ is $(1 - 2\beta_\epsilon)$. Note that, when $n$ is large, $\epsilon$ goes to zero, and $\beta_\epsilon \le 2\rho(1 - \rho)$, for $r = 1$.

If $\min_{f \in \mathcal{F}} R_{\ell,\mathcal{D}}(f)$ is sufficiently small, VC dimension is finite, and $\rho$ is bounded away from $1/2$ then for $n = O(m\log(m)/r)$, we get $\epsilon_1$ to be sufficiently small. Therefore, for any redundancy $r$, error in confusion matrix estimation is small when the number of training examples is sufficiently large. Hence, for $N$ large enough, using Equation (9) and the bound on $\beta_\epsilon$, we get that for fixed total annotation budget, the optimal choice of redundancy $r$ is 1 when the worker quality $(1 - \rho)$ is above a threshold. In particular, if $(1 - \rho) \ge 0.825$ then label once is the optimal strategy. However, in experiments we observe that with our algorithm the choice of $r = 1$ is optimal even for much smaller values of worker quality.

## 5 EXPERIMENTS

We experimentally investigate our algorithm, MBEM, on multiple large datasets. On CIFAR-10 (Krizhevsky & Hinton, 2009) and ImageNet (Deng et al., 2009), we draw noisy labels from synthetic worker models. We confirm our results on multiple worker models. On the MS-COCO dataset (Lin et al., 2014b), we accessed the real raw data that was used to produce this annotation. We compare MBEM against the following baselines:

- **MV**: First aggregate labels by performing a majority vote, then train the model.

- **weighted-MV**: Model learned using weighted loss function with weights set by majority vote.

- **EM**: First aggregate labels using EM. Then train model in the standard fashion. (Dawid & Skene, 1979)

- **weighted-EM**: Model learned using weighted loss function with weights set by standard EM.

- **oracle weighted EM**: This model is learned by minimizing $\ell_\pi$, using the true confusion matrices.

- **oracle correctly labeled**: This baseline is trained using the standard loss function $\ell$ but only using those training examples for which at least one of the $r$ workers has given the true label.

Note that oracle models cannot be deployed in practice. We show them to build understanding only. In the plots, the dashed lines correspond to MV and EM algorithm. The black dashed-dotted line shows generalization error if the model is trained using ground truth labels on all the training examples. For experiments with synthetic noisy workers, we consider two models of worker skill:

- **hammer-spammer:** Each worker is either a *hammer* (always correct) with probability $\gamma$ or a *spammer* (chooses labels uniformly at random).

- **class-wise hammer-spammer:** Each worker can be a hammer for some subset of classes and a spammer for the others. The confusion matrix in this case has two types of rows: (a) hammer class: row with all off-diagonal elements being 0. (b) spammer class: row with all elements being $1/|\mathcal{K}|$. A worker is a hammer for any class $k \in \mathcal{K}$ with probability $\gamma$.

We sample $m$ confusion matrices $\{\pi^{(a)}\}_{a \in [m]}$ according to the given worker skill distribution for a given $\gamma$. We assign $r$ workers uniformly at random to each example. Given the ground truth labels, we generate noisy labels according to the probabilities given in a worker's confusion matrix, using Equation (1). While our synthetic workers are sampled from these specific worker skill models, our algorithms do not use this information to estimate the confusion matrices. A Python implementation of the MBEM algorithm is available for download at https://github.com/khetan2/MBEM.

**CIFAR-10** This dataset has a total of 60K images belonging to 10 different classes where each class is represented by an equal number of images. We use 50K images for training the model and 10K images for testing. We use the ground truth labels to generate noisy labels from synthetic workers. We choose $m = 100$, and for each worker, sample confusion matrix of size $10 \times 10$ according to the worker skill distribution. All our experiments are carried out with a 20-layer ResNet

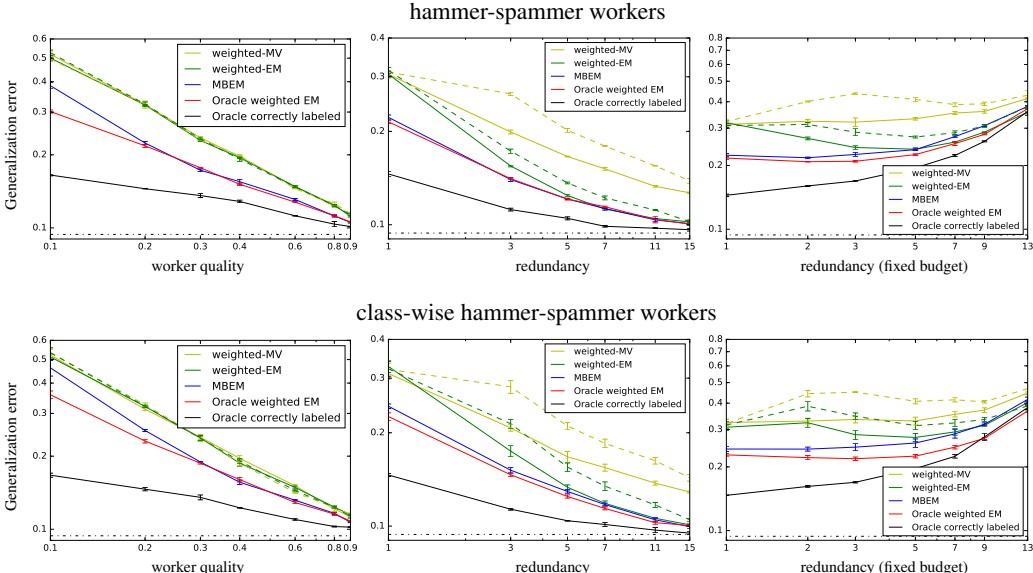

Figure 1: Plots for CIFAR-10. Line colors- black: oracle correctly labeled, red: oracle weighted EM, blue: MBEM, green: weighted EM, yellow: weighted MV.

which achieves an accuracy of $91.5\%$. With the larger ResNet-200, we can obtain a higher accuracy of $93.5\%$ but to save training time we restrict our attention to ResNet-20. We run MBEM 1 for $T = 2$ rounds. We assume that the prior distribution $\widehat{q}$ is uniform. We report mean accuracy of 5 runs and its standard error for all the experiments.

Figure 1 shows plots for CIFAR-10 dataset under various settings. The three plots in the first row correspond to "hammer-spammer" worker skill distribution and the plots in the second row correspond to "class-wise hammer-spammer" distribution. In the first plot, we fix redundancy $r = 1$, and plot generalization error of the model for varying hammer probability $\gamma$. MBEM significantly outperforms all baselines and closely matches the Oracle weighted EM. This implies MBEM recovers worker confusion matrices accurately even when we have only one label per example. When there is only one label per example, MV, weighted-MV, EM, and weighted-EM all reduce learning with the standard loss function $\ell$.

In the second plot, we fix hammer probability $\gamma = 0.2$, and vary redundancy $r$. This plot shows that weighted-MV and weighted-EM perform significantly better than MV and EM and confirms that our approach of weighing the loss function with posterior probability is effective. MBEM performs much better than weighted-EM at small redundancy, demonstrating the effect of our bootstrapping idea. However, when redundancy is large, EM works as good as MBEM.

In the third plot, we show that when the total annotation budget is fixed, it is optimal to collect one label per example for as many examples as possible. We fixed hammer probability $\gamma = 0.2$. Here, when redundancy is increased from 1 to 2, the number of of available training examples is reduced by 50%, and so on. Performance of weighted-EM improves when redundancy is increased from 1 to 5, showing that with the standard EM algorithm it might be better to collect redundant annotations for fewer example (as it leads to better estimation of worker qualities) than to singly annotate more examples. However, MBEM always performs better than the standard EM algorithm, achieving lowest generalization error with many singly annotated examples. Unlike standard EM, MBEM can estimate worker qualities even with singly annotated examples by comparing them with model predictions. This corroborates our theoretical result that label-once is the optimal strategy when worker quality is above a threshold. The plots corresponding to *class-wise hammer-spammer* workers follow the same trend. Estimation of confusion matrices in this setting is difficult and hence the gap between MBEM and the baselines is less pronounced.

**ImageNet** The ImageNet-1K dataset contains 1.2M training examples and 50K validation examples. We divide test set in two parts: 10K for validation and 40K for test. Each example belongs to one of the possible 1000 classes. We implement our algorithms using a ResNet-18 that achieves top-

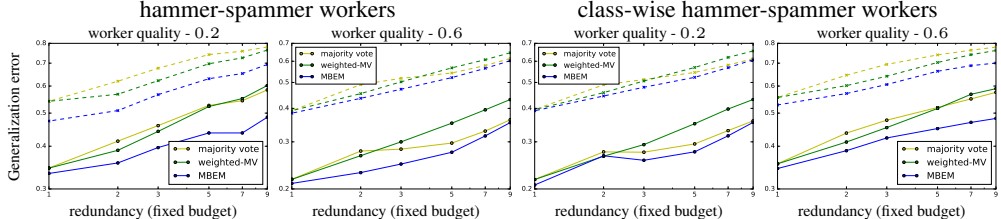

Figure 2: Plots for ImageNet. Solid lines represent top-5 error, dashed-lines represent top-1 error. Line colors- blue: MBEM, green: weighted majority vote, yellow: majority vote

| Approach | F1 score |
|---|---|
| majority vote | 0.433 |
| EM | 0.447 |
| MBEM | 0.451 |
| ground truth labels | 0.512 |

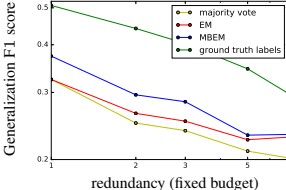

Figure 3: Results on raw MS-COCO annotations.

1 accuracy of 69.5% and top-5 accuracy of 89% on ground truth labels. We use $m = 1000$ simulated workers. Although in general, a worker can mislabel an example to one of the 1000 possible classes, our simulated workers mislabel an example to only one of the 10 possible classes. This captures the intuition that even with a larger number of classes, perhaps only a small number are easily confused for each other. Therefore, each workers' confusion matrix is of size $10 \times 10$. Note that without this assumption, there is little hope of estimating a $1000 \times 1000$ confusion matrix for each worker by collecting only approximately 1200 noisy labels from a worker. The rest of the settings are the same as in our CIFAR-10 experiments. In Figure 2, we fix total annotation budget to be 1.2M and vary redundancy from 1 to 9. When redundancy is 9, we have only $(1.2/9)$M training examples, each labeled by 9 workers. MBEM outperforms baselines in each of the plots, achieving the minimum generalization error with many singly annotated training examples.

**MS-COCO** These experiments use the real raw annotations collected when MS-COCO was crowdsourced. Each image in the dataset has multiple objects (approximately 3 on average). For validation set images (out of 40K), labels were collected from 9 workers on average. Each worker marks which out of the 80 possible objects are present. However, on many examples workers disagree. These annotations were collected to label bounding boxes but we ask a different question: what is the best way to learn a model to perform multi-object classification, using these noisy annotations. We use 35K images for training the model and 1K for validation and 4K for testing. We use raw noisy annotations for training the model and the final MS-COCO annotations as the ground truth for the validation and test set. We use ResNet-98 deep learning model and train independent binary classifier for each of the 80 object classes. Table in Figure 3 shows generalization F1 score of four different algorithms: majority vote, EM, MBEM using all 9 noisy annotations on each of the training examples, and a model trained using the ground truth labels. MBEM performs significantly better than the standard majority vote and slightly improves over EM. In the plot, we fix the total annotation budget to 35K. We vary redundancy from 1 to 7, and accordingly reduce the number of training examples to keep the total number of annotations fixed. When redundancy is $r < 9$ we select uniformly at random $r$ of the original 9 noisy annotations. Again, we find it best to singly annotate as many examples as possible when the total annotation budget is fixed. MBEM significantly outperforms majority voting and EM at small redundancy.

## 6 CONCLUSION

We introduced a new algorithm for learning from noisy crowd workers. We also presented a new theoretical and empirical demonstration of the insight that when examples are cheap and annotations expensive, it's better to label many examples once than to label few multiply when worker quality is

above a threshold. Many avenues seem ripe for future work. We are especially keen to incorporate our approach into active query schemes, choosing not only which examples to annotate, but which annotator to route them to based on our models current knowledge of both the data and the worker confusion matrices.

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

APPENDIX

## A    PROOF OF THEOREM 4.1

Assuming the prior on $Y$, distribution $q$, to be uniform, we change the notation for the modified loss function $\ell_{\widehat{\pi},\widehat{q}}$ to $\ell_{\widehat{\pi}}$. Observe that for binary classification, $Z^{(r)} \in \{\pm 1\}^r$. Let $\rho$ denote the posterior distribution of $Y$, Equation (5), when $q$ is uniform. Let $\tau$ denote the probability of observing an instance of $Z^{(r)}$ as a function of the latent true confusion matrices $\pi$, conditioned on the ground truth label $Y = y$.

$$\rho_{\widehat{\pi}}(y, Z^{(r)}, w^{(r)}) := \mathbb{P}_{\widehat{\pi}}[Y = y \mid Z^{(r)}; w^{(r)}], \qquad \tau_{\pi}(y, Z^{(r)}, w^{(r)}) := \mathbb{P}_{\pi}[Z^{(r)} \mid Y = y; w^{(r)}]. \quad (10)$$

Let $W$ denote the uniform distribution over a pool of $m$ workers, from which $nr$ workers are selected i.i.d. with replacement, and a batch of $r$ workers are assigned to each example $X_i$. We define the following quantities which play an important role in our analysis.

$$\beta_{\widehat{\pi}}(y) \quad := \quad \mathbb{E}_{w \sim W}\left[ \sum_{Z^{(r)} \in \{\pm 1\}^r} \rho_{\widehat{\pi}}(-y, Z^{(r)}, w^{(r)}) \tau_{\pi}(y, Z^{(r)}, w^{(r)}) \right]. \quad (11)$$

$$\beta_{\widehat{\pi}} \quad := \quad \mathbb{E}_{w \sim W}\left[ \max_{y \in \{\pm 1\}} \left\{ \sum_{Z^{(r)} \in \{\pm 1\}^r} \rho_{\widehat{\pi}}(-y, Z^{(r)}, w^{(r)}) \tau_{\pi}(y, Z^{(r)}, w^{(r)}) \right\} \right]. \quad (12)$$

$$\alpha(y) \quad := \quad \mathbb{E}_{w \sim W}\left[ \mathbb{P}_{\pi}[Z = -y \mid Y = y; w] \right]. \quad (13)$$

$$\alpha \quad := \quad \mathbb{E}_{w \sim W}\left[ \max_{y \in \{\pm 1\}} \mathbb{P}_{\pi}[Z = -y \mid Y = y; w] \right]. \quad (14)$$

For any given $\widehat{\pi}$ with $|\widehat{\pi}_{ks}^{(a)} - \pi_{ks}^{(a)}| \leq \epsilon$, for all $a \in [m]$, $k, s \in \mathcal{K}$, we can compute $\beta_{\epsilon}$ from the definition of $\beta_{\widehat{\pi}}$ such that $\beta_{\widehat{\pi}} \leq \beta_{\epsilon}$. For the special case described in Section 4.4, we have the following bound on $\beta_{\epsilon}$.

$$\beta_{\epsilon} \quad \leq \quad \sum_{u=0}^{r} \frac{(\rho + \epsilon)^{(r-u)}(1 - \rho - \epsilon)^u}{(\rho + \epsilon)^u(1 - \rho - \epsilon)^{(r-u)} + (\rho + \epsilon)^{(r-u)}(1 - \rho - \epsilon)^u} \binom{r}{u}(1 - \rho)^{r-u}\rho^u \quad (15)$$

$$= \quad (\rho + \epsilon)^r \sum_{u=0}^{r} \binom{r}{u}\left( \left(\frac{\rho + \epsilon}{1 - \rho - \epsilon}\right)^u + \left(\frac{\rho + \epsilon}{1 - \rho - \epsilon}\right)^{r-u} \right)^{-1}. \quad (16)$$

It can easily be checked that $\beta_{\epsilon} \leq (\rho + \epsilon)^r \sum_{u=0}^{\lceil r/2 \rceil} \binom{r}{u}(1 - \rho - \epsilon)^u(\rho + \epsilon)^{-u}$.

We present two lemma that analyze the two alternative steps of our algorithm. The following lemma gives a bound on the excess risk of function $\widehat{f}$ learnt by minimizing the modified loss function $\ell_{\widehat{\pi}}$.

**Lemma A.1.** *Under the assumptions of Theorem 4.1, the excess risk of function $\widehat{f}$ in Equation (6), computed with posterior distribution $\mathbb{P}_{\widehat{\pi}}$ (5) using $n$ training examples is bounded by*

$$R_{\ell,D}(\widehat{f}) - \min_{f \in \mathcal{F}} R_{\ell,D}(f) \quad \leq \quad \frac{C}{1 - 2\beta_{\widehat{\pi}}}\left( \sqrt{\frac{V}{n}} + \sqrt{\frac{\log(1/\delta_1)}{n}} \right), \quad (17)$$

*with probability at least $1 - \delta_1$, where $C$ is a universal constant. When $\mathbb{P}_{\widehat{\pi}}$ is computed using majority vote, while initializing the iterative Algorithm 1, the above bound holds with $\beta_{\widehat{\pi}}$ replaced by $\alpha$.*

The following lemma gives an $\ell_{\infty}$ norm bound on confusion matrices $\widehat{\pi}$ estimated using model prediction $\widehat{f}(X)$ as the ground truth labels. In the analysis, we assume fresh samples are used for estimating confusion matrices in step 3, Algorithm 1. Therefore the function $\widehat{f}$ is independent of the samples $X_i$'s on which $\widehat{\pi}$ is estimated. Let $K = |\mathcal{K}|$.

**Lemma A.2.** *Under the assumptions of Theorem 4.1, $\ell_\infty$ error in estimated confusion matrices $\widehat{\pi}$ as computed in Equation (7), using $n$ samples and a predictor function $\widehat{f}$ with risk $R_{\ell,\mathcal{D}} \leq \delta$, is bounded by*

$$\left| \widehat{\pi}_{ks}^{(a)} - \pi_{ks}^{(a)} \right| \;\leq\; \frac{2\delta + 16\sqrt{m\log(4mK^2\delta_1)/(nr)}}{1/K - \delta - 8\sqrt{m\log(4mK^2/\delta_1)/(nr)}}\,, \qquad \forall\, a \in [m],\; k,s \in \mathcal{K}\,, \quad (18)$$

*with probability at least $1 - \delta_1$.*

First we apply Lemma A.1 with $\mathbb{P}_{\widehat{\pi}}$ computed using majority vote. We get a bound on the risk of function $\widehat{f}$ computed in the first round. With this $\widehat{f}$, we apply Lemma A.2. When $n$ is sufficiently large such that Equation (8) holds, the denominator in Equation (18), $1/K - \delta - 8\sqrt{m\log(4mK^2/\delta_1)/(nr)} \geq 1/8$. Therefore, in the first round, the error in confusion matrix estimation is bounded by $\epsilon$, which is defined in the Theorem.

For the second round: we apply Lemma A.1 with $\mathbb{P}_{\widehat{\pi}}$ computed as the posterior distribution (5). Where $\ell_\infty$ error in $\widehat{\pi}$ is bounded by $\epsilon$. This gives the desired bound in (9). With this $\widehat{f}$, we apply Lemma A.2 and obtain $\ell_\infty$ error in $\widehat{\pi}$ bounded by $\epsilon_1$, which is defined in the Theorem.

For the given probability of error $\delta$ in the Theorem, we chose $\delta_1$ in both the lemma to be $\delta/4$ such that with union bound we get the desired probability of $\delta$.

## A.1 Proof of Lemma A.1

Let $f^* := \arg\min_{f \in \mathcal{F}} R_{\ell,\mathcal{D}}(f)$. Let's denote the distribution of $(X, Z^{(r)}, w^{(r)})$ by $\mathcal{D}_{W,\pi,r}$. For ease of notation, we denote $\mathcal{D}_{W,\pi,r}$ by $\mathcal{D}_\pi$. Similar to $R_{\ell,\mathcal{D}}$, risk of decision function $f$ with respect to the modified loss function $\ell_{\widehat{\pi}}$ is characterized by the following quantities:

1. $\ell_{\widehat{\pi}}$-risk under $\mathcal{D}_\pi$: $R_{\ell_{\widehat{\pi}},\mathcal{D}_\pi}(f) := \mathbb{E}_{(X,Z^{(r)},w^{(r)})\sim\mathcal{D}_\pi}\left[\ell_{\widehat{\pi}}(f(X), Z^{(r)}, w^{(r)})\right]$.
2. Empirical $\ell_{\widehat{\pi}}$-risk on samples: $\widehat{R}_{\ell_{\widehat{\pi}},\mathcal{D}_\pi}(f) := \frac{1}{n}\sum_{i=1}^n \ell_{\widehat{\pi}}(f(X_i), Z_i^{(r)}, w_i^{(r)})$.

With the above definitions, we have the following,

$$
\begin{aligned}
& R_{\ell,\mathcal{D}}(\widehat{f}) - R_{\ell,\mathcal{D}}(f^*) \\
={}& R_{\ell_{\widehat{\pi}},\mathcal{D}_\pi}(\widehat{f}) - R_{\ell_{\widehat{\pi}},\mathcal{D}_\pi}(f^*) + \left(R_{\ell,\mathcal{D}}(\widehat{f}) - R_{\ell_{\widehat{\pi}},\mathcal{D}_\pi}(\widehat{f})\right) - (R_{\ell,\mathcal{D}}(f^*) - R_{\ell_{\widehat{\pi}},\mathcal{D}_\pi}(f^*)) \\
\leq{}& R_{\ell_{\widehat{\pi}},\mathcal{D}_\pi}(\widehat{f}) - R_{\ell_{\widehat{\pi}},\mathcal{D}_\pi}(f^*) + 2\beta_{\widehat{\pi}}\left(R_{\ell,\mathcal{D}}(\widehat{f}) - R_{\ell,\mathcal{D}}(f^*)\right) & (19) \\
={}& \widehat{R}_{\ell_{\widehat{\pi}},\mathcal{D}_\pi}(\widehat{f}) - \widehat{R}_{\ell_{\widehat{\pi}},\mathcal{D}_\pi}(f^*) + \left(R_{\ell_{\widehat{\pi}},\mathcal{D}_\pi}(\widehat{f}) - \widehat{R}_{\ell_{\widehat{\pi}},\mathcal{D}_\pi}(\widehat{f})\right) + \left(\widehat{R}_{\ell_{\widehat{\pi}},\mathcal{D}_\pi}(f^*) - R_{\ell_{\widehat{\pi}},\mathcal{D}_\pi}(f^*)\right) \\
& + 2\beta_{\widehat{\pi}}\left(R_{\ell,\mathcal{D}}(\widehat{f}) - R_{\ell,\mathcal{D}}(f^*)\right) \\
\leq{}& 2\max_{f \in \mathcal{F}}\left|\widehat{R}_{\ell_{\widehat{\pi}},\mathcal{D}_\pi}(f) - R_{\ell_{\widehat{\pi}},\mathcal{D}_\pi}(f)\right| + 2\beta_{\widehat{\pi}}\left(R_{\ell,\mathcal{D}}(\widehat{f}) - R_{\ell,\mathcal{D}}(f^*)\right) & (20) \\
\leq{}& C\left(\sqrt{\frac{V}{n}} + \sqrt{\frac{\log(1/\delta)}{n}}\right) + 2\beta_{\widehat{\pi}}\left(R_{\ell,\mathcal{D}}(\widehat{f}) - R_{\ell,\mathcal{D}}(f^*)\right)\,, & (21)
\end{aligned}
$$

where (19) follows from Equation (24). (20) follows from the fact that $\widehat{f}$ is the minimizer of $\widehat{R}_{\ell_{\widehat{\pi}},\mathcal{D}_\pi}$ as computed in (6). (21) follows from the basic excess-risk bound. $V$ is the VC dimension of hypothesis class $\mathcal{F}$, and $C$ is a universal constant.

Following shows the inequality used in Equation (19). For binary classification, we denote the two classes by $Y, -Y$.

$$
\begin{aligned}
={}& R_{\ell,\mathcal{D}}(\widehat{f}) - R_{\ell_{\widehat{\pi}},\mathcal{D}_\pi}(\widehat{f}) - (R_{\ell,\mathcal{D}}(f^*) - R_{\ell_{\widehat{\pi}},\mathcal{D}_\pi}(f^*)) \\
={}& \mathbb{E}_{(X,Y)\sim\mathcal{D}}\left[\beta_{\widehat{\pi}}(Y)\left(\left(\ell(\widehat{f}(X), Y) - \ell(f^*(X), Y)\right) - \left(\ell(\widehat{f}(X), -Y) - \ell(f^*(X), -Y)\right)\right)\right] & (22) \\
={}& 2\mathbb{E}_{(X,Y)\sim\mathcal{D}}\left[\beta_{\widehat{\pi}}(Y)\left(\ell(\widehat{f}(X), Y) - \ell(f^*(X), Y)\right)\right] & (23) \\
\leq{}& 2\beta_{\widehat{\pi}}\left(R_{\ell,\mathcal{D}}(\widehat{f}) - R_{\ell,\mathcal{D}}(f^*)\right)\,, & (24)
\end{aligned}
$$

where (22) follows from Equation (26). (23) follows from the fact that for 0-1 loss function $\ell(f(X), Y) + \ell(f(X), -Y) = 1$. (24) follows from the definition of $\beta_{\widehat{\pi}}$ defined in Equation (12). When $\ell_{\widehat{\pi}}$ is computed using weighted majority vote of the workers then (24) holds with $\beta_{\widehat{\pi}}$ replaced by $\alpha$. $\alpha$ is defined in (14).

Following shows the equality used in Equation (22). Using the notations $\rho_{\widehat{\pi}}$ and $\tau_{\pi}$, in the following, for any function $f \in \mathcal{F}$, we compute the excess risk due to the unbiasedness of the modified loss function $\ell_{\widehat{\pi}}$.

$$
\begin{aligned}
& R_{\ell,\mathcal{D}}(f) - R_{\ell_{\widehat{\pi}}, \mathcal{D}_{\pi}}(f) \\
=~ & \mathbb{E}_{(X,Y) \sim \mathcal{D}}\left[\ell(f(X), Y)\right] - \mathbb{E}_{(X, Z^{(r)}, w^{(r)}) \sim \mathcal{D}_{\pi}}\left[\ell_{\widehat{\pi}}(f(X), Z^{(r)}, w^{(r)})\right] \\
=~ & \mathbb{E}_{(X,Y) \sim \mathcal{D}}\left[\ell(f(X), Y)\right] \\
& - \mathbb{E}_{(X,Y,w^{(r)}) \sim \mathcal{D}_{\pi}}\Bigg[ \sum_{Z^{(r)} \in \{\pm 1\}^r} \Big((1 - \rho_{\widehat{\pi}}(-Y, Z^{(r)}, w^{(r)}))\ell(f(X), Y) \\
& \qquad\qquad + \rho_{\widehat{\pi}}(-Y, Z^{(r)}, w^{(r)})\ell(f(X), -Y)\Big)\tau_{\pi}(Y, Z^{(r)}, w^{(r)})\Bigg] \\
=~ & \mathbb{E}_{(X,Y) \sim \mathcal{D}}\left[\beta_{\widehat{\pi}}(Y)\left(\ell(f(X), Y) - \ell(f(X), -Y)\right)\right],
\end{aligned}
\tag{25}
$$

$$
\tag{26}
$$

where $\beta_{\widehat{\pi}}(Y)$ is defined in (11). Where (25) follows from the definition of $\ell_{\widehat{\pi}}$ given in Equation (4). Observe that when $\ell_{\widehat{\pi}}$ is computed using weighted majority vote of the workers then Equation (26) holds with $\beta_{\widehat{\pi}}(Y)$ replaced by $\alpha(y)$. $\alpha(y)$ is defined in (13).

## A.2 PROOF OF LEMMA A.2

Recall that we have

$$
\widehat{\pi}_{ks}^{(a)} = \frac{\sum_{i=1}^{n} \sum_{j=1}^{r} \mathbb{I}[w_{ij} = a]\mathbb{I}[t_i = k]\mathbb{I}[Z_{ij} = s]}{\sum_{i=1}^{n} \sum_{j=1}^{r} \mathbb{I}[w_{ij} = a]\mathbb{I}[t_i = k]}
\tag{27}
$$

Let $t_i$ denote $\widehat{f}(X_i)$. By the definition of risk, for any $k \in \mathcal{K}$, we have

$$
\mathbb{P}\Big[\big|\mathbb{I}[Y_i = k] - \mathbb{I}[t_i = k]\big| = 1\Big] = \delta.
$$

Let $|\mathcal{K}| = K$. Define, for fixed $a \in [m]$, and $k, s \in \mathcal{K}$,

$$
A := \sum_{i=1}^{n} \sum_{j=1}^{r} \mathbb{I}[w_{ij} = a]\mathbb{I}[t_i = k]\mathbb{I}[Z_{ij} = s], \qquad \bar{A} := \frac{nr\pi_{ks}}{mK}
\tag{28}
$$

$$
B := \sum_{i=1}^{n} \sum_{j=1}^{r} \mathbb{I}[w_{ij} = a]\mathbb{I}[t_i = k], \qquad \bar{B} := \frac{nr}{mK}
\tag{29}
$$

$$
C := \sum_{i=1}^{n} \sum_{j=1}^{r} \mathbb{I}[w_{ij} = a]\big|\mathbb{I}[Y_i = k] - \mathbb{I}[t_i = k]\big|, \qquad \bar{C} := \frac{nr\delta}{m},
\tag{30}
$$

$$
D := \sum_{i=1}^{n} \sum_{j=1}^{r} \mathbb{I}[w_{ij} = a]\mathbb{I}[Y_i = k]\mathbb{I}[Z_{ij} = s],
\tag{31}
$$

$$
E := \sum_{i=1}^{n} \sum_{j=1}^{r} \mathbb{I}[w_{ij} = a]\mathbb{I}[Y_i = k].
\tag{32}
$$

Note that $A, B, C, D, E$ depend upon $a \in [m]$, $k, s \in \mathcal{K}$. However, for ease of notations, we have not included the subscripts. We have,

$$
\begin{aligned}
\left|\widehat{\pi}_{ks}^{(a)} - \pi_{ks}^{(a)}\right| = \frac{A - B\pi_{ks}}{B} &= \frac{|(A - \bar{A}) - (B - \bar{B})\pi_{ks}|}{|\bar{B} + (B - \bar{B})|} \\
&\leq \frac{|A - \bar{A}| + |(B - \bar{B})|\pi_{ks}}{|\bar{B}| - |B - \bar{B}|}
\end{aligned}
\tag{33}
$$

Now, we have,

$$
\begin{aligned}
|A - \bar{A}| &\leq |A - D| + |D - \bar{A}| \\
&\leq C + |D - \bar{A}|.
\end{aligned}
\tag{34}
$$

We have,

$$
\begin{aligned}
|B - \bar{B}| &\leq |B - E| + |E - \bar{B}| \\
&\leq C + |E - \bar{B}|
\end{aligned}
\tag{35}
$$

Observe that $C$ is a sum of $nr$ i.i.d. Bernoulli random variables with mean $\delta/m$. Using Chernoff bound we get that

$$
C \leq \frac{nr\delta}{m} + \sqrt{\frac{3nr\delta \log(2mK/\delta_1)}{m}},
\tag{36}
$$

for all $a \in [m]$, and $k \in \mathcal{K}$ with probability at least $1 - \delta_1$. Similarly, $D$ is a sum of $nr$ i.i.d. Bernoulli random variables with mean $\pi_{ks}/(mk)$. Again, using Chernoff bound we get that

$$
|D - \bar{A}| \leq \sqrt{\frac{3nr\pi_{ks} \log(2mK^2/\delta_1)}{mK}},
\tag{37}
$$

for all $a \in [m]$, $k, s \in \mathcal{K}$ with probability at least $1 - \delta_1$. From the bound on $|D - \bar{A}|$, it follows that

$$
|E - \bar{B}| \leq \sqrt{\frac{3nr \log(2mK^2/\delta_1)}{m}}
\tag{38}
$$

Collecting Equations (33)-(38), we have for all $a \in [m]$, $k, s \in \mathcal{K}$

$$
\left| \widehat{\pi}_{ks}^{(a)} - \pi_{ks}^{(a)} \right| \leq \frac{2\delta + 16\sqrt{m \log(2mK^2\delta_1/(nr))}}{1/K - \delta - 8\sqrt{m \log(2mK^2/\delta_1)/(nr)}},
\tag{39}
$$

with probability at least $1 - 2\delta_1$.

