# OpenReview forum: "Learning From Noisy Singly-labeled Data"
_ICLR.cc/2018/Conference — Accept (Poster)_

### Official Review · AnonReviewer2 · 2017-11-25
**Well-written paper with extensive experiments, but missing a realistic non-simulated experiment, as well as some comparisons and related work.**

**Rating:** 7
**Confidence:** 4

**Review:**

This paper proposes a method for learning from noisy labels, particularly focusing on the case when data isn't redundantly labeled (i.e. the same sample isn't labeled by multiple non-expert annotators). The authors provide both theoretical and experimental validation of their idea.

Pros:
+ The paper is generally very clearly written. The motivation, notation, and method are clear.
+ Plentiful experiments against relevant baselines are included, validating both the no-redundancy and plentiful redundancy cases.
+ The approach is a novel twist on an existing method for learning from noisy data.

Cons:
- All experiments use simulated workers; this is probably common but still not very convincing.
- The authors missed an important related work which studies the same problem and comes up with a similar conclusion: Lin, Mausam, and Weld. "To re (label), or not to re (label)." HCOMP 2014.
- The authors should have compared their approach to the "base" approach of Natarajan et al.
- It seems too simplistic too assume all workers are either hammers or spammers; the interesting cases are when annotators are neither of these.
- The ResNet used for each experiment is different, and there is no explanation of the choice of architecture.

Questions:
- How would the model need to change to account for example difficulty?
- Why are Joulin 2016, Krause 2016 not relevant?
- Best to clarify what the weights in the weighted sum of Natarajan are.
- "large training error on wrongly labeled examples" -- how do we know they are wrongly labeled, i.e. do we have a ground truth available apart from the crowdsourced labels? Where does this ground truth come from?
- Not clear what "Ensure" means in the algorithm description.
- In Sec. 4.4, why is it important that the samples are fresh?

---

> ### Author Response · Authors · 2017-12-30
> **Response to AnonReviewer2**
>
> Replies to each point follow:
>
> 1. Re “All experiments use simulated workers; this is probably common but still not very convincing.”
>
> Please note that in experiments on MSCOCO, we procured the real noisy labels from the raw data. See in abstract: “Experiments conducted on … and MSCOCO (using the real crowdsourced labels)...”
>
> 2. Re “The authors missed an important related work which studies the same problem and comes up with a similar conclusion: Lin, Mausam, and Weld. "To re (label), or not to re (label)." HCOMP 2014.”
>
> We agree that this is one of the most relevant works. Note that we cited this work along with their 2016 paper along similar lines “Re-active learning: Active learning with relabeling.”. Also, note that unlike ours, their work does not use predictions of the supervised learning algorithm to estimate the true labels.
>
> 3. Re “The authors should have compared their approach to the "base" approach of Natarajan et al.”
>
> Their approach is designed for the binary classification setting when all the workers are identical. We study the multi-class classification setting where workers have varying qualities.
>
> 4. Re “It seems too simplistic to assume all workers are either hammers or spammers; the interesting cases are when annotators are neither of these.”
>
> We agree and point out (1) that we considered two other worker models. In synthetic dataset, e.g. we consider class-wise hammer spammer, where each worker is hammer for some of the classes and spammer for the other classes. (2) We report experiments on MSCOCO with labels collected by real workers.
>
> 5. Re “The ResNet used for each experiment is different, and there is no explanation of the choice of architecture.”
>
> For simulated worker experiments on CIFAR10 and ImageNet, we used the fewest possible layers. These choices are dictated by the ResNet implementation that we used “https://github.com/tornadomeet/ResNet/”. Smaller ResNet architectures save training time, enabling us to perform experiments on more baseline algorithms, worker noise models, and levels of redundancy.
>
> For MSCOCO experiments, we used a 98-layer ResNet because this is a relatively small dataset. Also, we did not have various experiments to run for different worker noise models here.
>
> 6. Re “How would the model need to change to account for example difficulty? ”
>
> When we include example difficulty in the model, there are three sets of latent parameters to be estimated: worker qualities, example difficulties and the true labels. A standard approach to learn these parameters is to use alternating maximum likelihood estimation where we initialize the two sets of parameters and estimate the third one and iterate over. In our algorithm, we would need to estimate example difficulties by maximizing the likelihood of the observed data given the intermediate estimate of worker qualities and the labeling function.
>
> 7. Re “Why are Joulin 2016, Krause 2016 not relevant?”
>
> Two important differences between these works & our setting: a) they have only one label per example - no redundancy. b) they do not aim to estimate worker qualities.
>
> 8. Re “Best to clarify what the weights in the weighted sum of Natarajan are.”
>
> Update: We have provided the weights in Natarajan et al in the revised draft in the first paragraph of Section 4.1- “Learning with noisy labels”.
>
> 9. Re “"large training error on wrongly labeled examples" -- how do we know they are wrongly labeled...?”
> You are correct that we do not know which examples are wrongly labeled and we do not have ground truth available apart from the crowdsourced labels. We would humbly point out that the statement "large training error on wrongly labeled examples" is not a part of our algorithm. The purpose of the statement is to justify why comparing worker responses to the model prediction would give a good estimation of the worker qualities. It is further elaborated in the text below the line "large training error on wrongly labeled examples".
>
> 10.  Re “Not clear what "Ensure" means in the algorithm description.”
>
> In the algorithmic package, “Input” and “Output” are expressed with “Require” and “Ensure” respectively. So “Ensure” just means the output of the algorithm. We agree that “Input” and “Output” are clearer and modified the latest version to use these terms.
>
> 11. Re “In Sec. 4.4, why is it important that the samples are fresh?”
>
> As we mention in the paper, fresh samples are required for the analysis to hold. It allows the estimated worker qualities and the predictor function learned in each step to be independent of each other which is required for the Theorem 4.1 to hold. We point out that practically, fresh samples are not required for the algorithm to succeed, and in our implementation we do not use fresh samples in each round.

---

> > ### Comment · AnonReviewer2 · 2018-01-13
> > **thanks!**
> >
> > I read the rebuttal and I am leaning positive. I am going to update my score.

---

### Official Review · AnonReviewer3 · 2017-11-27
**A good paper with relatively pool experiments.**

**Rating:** 6
**Confidence:** 3

**Review:**

This paper focuses on the learning-from-crowds problem when there is only one (or very few) noisy label per item. The main framework is based on the Dawid-Skene model. By jointly update the classifier weights and the confusion matrices of workers, the predictions of the classifier can help on the estimation problem with rare crowdsourced labels. The paper discusses the influence of the label redundancy both theoretically and empirically. Results show that with a fixed budget, it’s better to label many examples once rather than fewer examples multiple times.

The model and algorithm in this paper are simple and straightforward. However, I like the motivation of this paper and the discussion about the relationship between training efficiency and label redundancy. The problem of label aggregation with low redundancy is common in practice but hardly be formally analyzed and discussed. The conclusion that labeling more examples once is better can inspire other researchers to find more efficient ways to improve crowdsourcing.

About the technique details, this paper is clearly written, but some experimental comparisons and claims are not very convincing. Here I list some of my questions:
+About the MBEM algorithm, it’s better to make clear the difference between MBEM and a standard EM. Will it always converge? What’s its objective?
+The setting of Theorem 4.1 seems too simple. Can the results be extended to more general settings, such as when workers are not identical?
+When n = O(m log m), the result that \epslon_1 is constant is counterintuitive, people usually think large redundancy r can bring benefits on estimation, can you explain more on this?
+During CIFAR-10 experiments when r=1, each example only have one label. For the baselines weighted-MV and weighted-EM, they can only be directly trained using the same noisy labels. So can you explain why their performance is slightly different in most settings? Is it due to the randomly chosen procedure of the noisy labels?
+For ImageNet and MS-COCO experiments with a fixed budget, you reduced the training set when increasing the redundancy, which is unfair. The reduction of performance could mainly cause by seeing fewer raw images, but not the labels. It’s better to train some semi-supervised model to make the settings more comparable.

---

> ### Author Response · Authors · 2017-12-30
> **Response to AnonReviewer3**
>
> Thanks for the clear review and actionable recommendations. We have modified the draft per your feedback and reply to each point below:
> 1. Re: “What’s [MBEM’s] objective?”: Thanks for spotting this oversight. The objective for MBEM is the maximum likelihood estimation of latent parameters under the Dawid-Skene model, where the true labels are replaced by the model predictions. We have added this to the revised draft in Section 4, Algorithm. Yes, the MBEM will converge under mild conditions when the worker quality is above a threshold and number of training examples is sufficiently large.
> 2. Re: “Can the results be extended to more general settings, such as when workers are not identical?
> Please note that the Theorem 4.1 includes the scenario when the workers are not identical. The two critical quantities $\alpha$ and $\beta$ that capture the average worker quality in the Theorem are defined for a general setting when the workers are not identical in the appendix. For simplicity and to illustrate the main idea of the theorem in the main paper we have defined them for the particular setting when all the workers are identical.
>
> 3. Re: “When n = O(m log m), the result that \epslon_1 is constant is counterintuitive, people usually think large redundancy r can bring benefits on estimation, can you explain more on this?”
> The expression O(m log m) hides redundancy r as a constant. In the revised draft, we have modified the statement to “when n = O((m log m)/r) the epsilon_1 is sufficiently small.” That is if the redundancy r is large the number of training examples n required for achieving epsilon_1 to be a small constant decreases.
>
> 4. Re “During CIFAR-10 experiments when r=1, each example only have one label. For the baselines weighted-MV and weighted-EM, they can only be directly trained using the same noisy labels. So can you explain why their performance is slightly different in most settings? Is it due to the randomly chosen procedure of the noisy labels? ”
>
> Yes, you are correct. When r =1, the baselines weighted-MV and weighted-EM can only be trained using the same noisy labels. Please note that R=1 is only in the left-most figures for CIFAR10 experiments for the two settings of hammer-spammer and class-wise hammer-spammer, respectively. In these plots, the lines for weighted-MV and weighted-EM are nearly identical. The negligible differences owe only to random worker assignment and random initialization of parameters. In rest of the four figures of CIFAR10 experiments, the redundancy r varies along the x-axis.
>
> 5. Re “For ImageNet and MS-COCO experiments with a fixed budget, you reduced the training set when increasing the redundancy, which is unfair. The reduction of performance could mainly cause by seeing fewer raw images, but not the labels. It’s better to train some semi-supervised model to make the settings more comparable.”
>
> We agree that in principle, the strongest baseline to prove our point that labeling once is optimal would allow the redundant labelers to make used of the unlabelled data in a semi-supervised fashion. We note that this does not directly fall out of our theory, which addresses the supervised case (see Theorem 4.1) and thus may be beyond the scope of this paper. We also note that many current semi-supervised algorithms, such as Ladder Networks, show most significant improvements when the ratio of unlabeled to labeled data is quite large, and that it is not clear how advantageous current semi-supervised algorithms would be at a redundancy level of say 3. While answering these questions conclusively is a non-trivial task and left for future work, we think that this is a great point and plan to investigate in the future how the utility of unlabeled data for semi-supervised learning may complicate the picture.

---

### Official Review · AnonReviewer1 · 2017-11-27
**Novel approach and a well written paper. Need more details on theoretical guarantees for multi-class settings and detailed analysis on real world data.**

**Rating:** 7
**Confidence:** 4

**Review:**

The authors proposed a supervised learning algorithm for modeling label and worker quality. Further utilize it to study one of the important problems in crowdsourcing - How much redundancy is required in crowdsourcing and whether low redundancy with abundant noise examples lead to better labels.

Overall the paper was well written. The motivation of the work is clearly explained and supported with relevant related work. The main contribution of the paper is in the bootstrapping algorithm which models the worker quality and labels in an iterative fashion. Though limited to binary classification, the paper proposed a theoretical framework extending the existing work on VC dimension to compute the upper bound on the risk. The authors also showed theoretically and empirically on synthetic data sets that the low redundancy and larger set of labels in crowdsourcing gives better results.

More detailed comments
1. Instead of considering multi-class classification as one-vs-all binary classification, can you extend the theoretical guarantee on the risk to multi-class set up like Softmax which is widely used in research nowadays.
2. Can you introduce the Risk -R in the paper before using it in Theorem 4.1
3. Is there any limit on how many examples each worker has to label? Can you comment more on how to pick that value in real-world settings? Just saying sufficiently many (Section 4.2) is not sufficient.
4. Under the experiments, different variations of Majority Vote, EM and Oracle correction were used as baselines. Can you cite the references and also add some existing state-of-the-art techniques mentioned in the related work section.
5. For the experiments on synthetic datasets, workers are randomly sampled with replacements. Were the scores reported based on average of multiple runs. If yes, can you please report the error bars.
6. For the MS-COCO, examples can you provide more detailed results as shown for synthetic datasets? Majority vote is a very weak baseline.

For the novel approach and the theoretical backing, I consider the paper to be a good one. The paper has scope for improvement.

---

> ### Author Response · Authors · 2017-12-30
> **Response to AnonReviewer1**
>
> Thanks for the thoughtful review and clearly enumerated critical points. We reply to each below:
>
> 1. We agree that it would be desirable to extend the theoretical guarantees to the multiclass-classification setting with cross-entropy loss and we plan to explore this question in future work. However, this extension is non-trivial under the current framework. Equation 22 in Lemma A.2 does not apply for cross-entropy loss and it is not obvious how to complete the guarantees without this result.
>
> 2. In the initial draft, we introduced the Risk -R in the problem formulation section. We’re grateful for the feedback that this was not obvious when you arrived at theorem 4.1 and we have modified the draft to remind the reader at this point.
>
> 3. Equation 7 in Theorem 4.1 states the condition on how many examples each worker has to label for the algorithm to succeed in estimating worker qualities. In particular, given m workers the algorithm needs to estimate O(m) latent parameters of their confusion matrices. From standard statistical analysis as reflected in Equation 7, we need O(m log m) independent observations to estimate O(m) parameters. Therefore, if we have n training examples, and redundancy is r then the total number of observations nr should satisfy: nr  > m log m. Hence, each worker has to label O(log m) examples for the algorithm to succeed.
>
> 4. We have compared our algorithm MBEM with four different algorithms and two oracle-based algorithms. Majority vote is a standard algorithm and Expectation Maximization (EM) is based on the classical Dawid Skene (1979) work, we have included reference to it in the revised draft. Weighted MV and weighted EM use a weighted loss function that is newly proposed in this work. The purpose of including these algorithms is to establish efficacy of weighted loss function over the standard loss function for noisy labels. Note that MBEM uses the weighted loss function in addition to the bootstrapping idea to estimate worker qualities.
>
> We  appreciate the request for a comparison against state-of-the-art techniques mentioned in the related work section. We are presently implementing the method from “Lean  Crowdsourcing” (Branson, Van Horn, Petrona 2017) as an additional baseline method and will add the results to the experiments section as soon as they are available (http://openaccess.thecvf.com/content_cvpr_2017/html/Branson_Lean_Crowdsourcing_Combining_CVPR_2017_paper.html).
>
> 5. Per your suggestions, in the new (current) draft we added the error bars for CIFAR10. In the initial draft, we were reporting averages across multiple runs for CIFAR10 and MSCOCO. For ImageNet, the experiments are too expensive, so we only execute one run.
>
> 6. We will add the results of the EM algorithm and weighted EM algorithm for MSCOCO experiment. We are also working presently on adding the method due to Branson et al. as a baseline to the MSCOCO experiment and will post results when available.

---

### Author Response · Authors · 2017-12-30
**General reply to all reviewers**

We would like to thank all of the reviewers for providing us with three clear and thoughtful reviews. We were encouraged both to see that the reviews were generally positive and that the recommendations were clear and actionable. We have acted on many of these recommendations and the current draft has been improved significantly. For example, in Section 4, the articulation of the objective for the MBEM algorithm is made explicit. Additionally we now report error bars for our CIFAR experiments. Additional experiments are underway and we will post these improvements as they become available. Please find specific replies to each review in the respective threads.

---

> ### Author Response · Authors · 2018-01-05
> **Added one more baseline algorithm to the MSCOCO experiments**
>
> We have added the results of the EM algorithm as an additional baseline to the MSCOCO experiments.

---

### Public Comment · ~Theodoros_Rekatsinas1 · 2018-01-07
**What about recent weak supervision methods?**

How does this paper compare to recent work by Ratner et al. (see Data programming at NIPS 2016)? Also how does this work compare to all the data fusion work in the database community? Please see https://arxiv.org/abs/1505.02463 and  https://dl.acm.org/citation.cfm?id=3035951.

While the authors might not be aware of the work done by the database community on data fusion algorithms they should have compared against the work on data programming and weak supervision https://hazyresearch.github.io/snorkel/

---

> ### Author Response · Authors · 2018-01-08
> **Differences with the "Data Programming: Creating Large Training Sets, Quickly (NIPS 2016)" paper.**
>
> Thanks for the comment, and for pointing to relevant works in the literature. While we are familiar with and value these papers, there are substantial differences between their work and ours. Below, we describe the differences between our work and the papers that you mentioned:
>
> Data Programming: Creating Large Training Sets, Quickly (NIPS 2016):
> We agree that this is a relevant prior-work and we will add it to our related works section. Please note that there are two critical differences between their algorithm and ours:
>
> (a) They propose to minimize the expected loss for each training example and each noisy label (after estimating the noise parameters). In contrast, we propose to minimize a weighted loss function for each training example. Our weights are the posterior probabilities of the true label given all the redundant noisy labels and the estimated noise parameters. If there is only one label per example then the two loss functions are same. However, for more than one label per example, the two loss functions are significantly different. Up to this difference, their work when applied in our setting reduces to one of our baseline algorithms, weighted-EM.
>
> (b) A major gain of our algorithm is in its iterative approach where we use model predictions to refine the estimation of noise parameters and learn a better model iteratively. This approach allows us to learn noise parameters (confusion matrices of workers) even when we collect only one label per example. There is no such iterative approach proposed in their work.
>
> Data fusion algorithms developed by database community:
> These algorithms are relevant to standard crowdsourcing algorithms which do not use features of the tasks to train a supervised learning algorithm. Their end goal is to establish the true label of the tasks given multiple noisy labels (by assessing noise rate of the different noisy labelers). On the other hand, in our paper and in the Data programming paper, the end goal is to train a classifier given the noisy labels.

---

### Author Response · Authors · 2018-02-20
**python implementation**

Following is the link to python code for implementing the MBEM algorithm presented in the paper. https://github.com/khetan2/MBEM

---

### Decision · Program_Chairs · 2018-01-29
**ICLR 2018 Conference Acceptance Decision**

**Decision:**

Accept (Poster)

**Comment:**

This paper provides an important discussion about the relationship between training efficiency and label redundancy. The updates to the paper will improve the paper further. Reviewers found the paper interesting, well written, and addresses and important problem.